# From pandemic response to portable population health: A formative evaluation of the Detroit mobile health unit program

**Phillip Levy**[1], **Erin McGlynn**[1]*, **Alex B. Hill**[1], **Liying Zhang**[2], **Steven J. Korzeniewski**[2],
**Bethany Foster**[1], **Jasmine Criswell**[3], **Caitlin O'Brien**[3], **Katee Dawood**[3], **Lauren Baird**[3],
**Charles J. Shanley**[4]

1 Department of Emergency Medicine, Wayne State University School of Medicine, Detroit, Michigan, United
States of America, 2 Department of Family Medicine and Public Health Sciences, Wayne State University
School of Medicine, Detroit, Michigan, United States of America, 3 Wayne Health, Wayne State University,
Detroit, Michigan, United States of America, 4 Department of Surgery, Wayne State University School of
Medicine, Detroit, Michigan, United States of America

* ekmcglynn@wayne.edu

UNITED STATES

**Data Availability Statement:** All relevant data are
within the paper and is also available publicly at
https://www.waynehealthcares.org/mobile-health-
unit/#statistics.

## Abstract

This article describes our experience developing a novel mobile health unit (MHU) program
in the Detroit, Michigan, metropolitan area. Our main objectives were to improve healthcare
accessibility, quality and equity in our community during the novel coronavirus pandemic.
While initially focused on SARS-CoV-2 testing, our program quickly evolved to include pre-
ventive health services. The MHU program began as a location-based SARS-CoV-2 testing
strategy coordinated with local and state public health agencies. Community needs moti-
vated further program expansion to include additional preventive healthcare and social ser-
vices. MHU deployment was targeted to disease "hotspots" based on publicly available
SARS-CoV-2 testing data and community-level information about social vulnerability. This
formative evaluation explores whether our MHU deployment strategy enabled us to reach
patients from communities with heightened social vulnerability as intended. From 3/20/20-3/
24/21, the Detroit MHU program reached a total of 32,523 people. The proportion of patients
who resided in communities with top quartile Centers for Disease Control and Prevention
Social Vulnerability Index rankings increased from 25% during location-based "drive-
through" SARS-CoV-2 testing (3/20/20-4/13/20) to 27% after pivoting to a mobile platform
(4/13/20-to-8/31/20; p = 0.01). The adoption of a data-driven deployment strategy resulted
in further improvement; 41% of the patients who sought MHU services from 9/1/20-to-3/24/
21 lived in vulnerable communities (Cochrane Armitage test for trend, p<0.001). Since 10/1/
21, 1,837 people received social service referrals and, as of 3/15/21, 4,603 were adminis-
tered at least one dose of COVID-19 vaccine. Our MHU program demonstrates the capacity
to provide needed healthcare and social services to difficult-to-reach populations from areas
with heightened social vulnerability. This model can be expanded to meet emerging pan-
demic needs, but it is also uniquely capable of improving health equity by addressing long-
standing gaps in primary care and social services in vulnerable communities.

**Funding:** Funding was supplied by donors and non-profit organizations including United Way for Southeastern Michigan, the Community Foundation of Southeast Michigan/Detroit Medical Center Foundation, the Ralph C. Wilson Foundation, Community Organized Relief Effort (CORE), DTE Energy Foundation, Blue Cross Blue Shield of Michigan, and the Cielo Foundation. Michigan Department of Health and Human Services (MDHHS) also collaborated and contributed funding to support further growth and extension of services. A CDC funded program (1817) with the MDHHS Heart Disease and Stroke Prevention Unit allowed for cardiometabolic risk factor screening. In addition, funding for the PHOENIX program was provided by the Michigan Health Endowment Fund and Delta Dental Michigan.

**Competing interests:** The authors have declared that no competing interests exist.

# Introduction

Difficult-to-reach populations from areas with heightened vulnerability related to socioeconomic status have suffered disproportionately from SARS-CoV-2 and its sequelae (i.e., COVID-19) [1–3]. Multiple community-level risk factors including poor access to healthcare and social services, household overcrowding [4, 5], increased reliance on public transportation [6] and other correlates of poverty (e.g., chronic cardiometabolic disease) appear to contribute to elevated risks [1–3]. Thus, there has been accelerated interest in alternative community-based healthcare strategies that target vulnerable populations.

Outreach into communities using vehicle-based platforms offers tremendous flexibility and enhanced capacity to help meet the needs of vulnerable populations. In comparison to temporary shelters and "pop up" healthcare clinics, mobile health units (MHUs) can adapt to evolving community needs more easily and they are readily accessible to people without transportation. Moreover, by bringing care directly to people in their neighborhoods, MHUs can help improve health equity by filling gaps in primary/preventive care and chronic disease management (e.g., by integrating with a rapidly expanding telehealth ecosystem).

This article describes our experience developing and deploying a fleet of five MHUs in the Detroit, Michigan, metropolitan area. First, recognizing the potential value of MHUs early on in the COVID-19 pandemic, we partnered with the Ford Motor Company to field-test a novel mobile SARS-CoV-2 testing platform. Second, we leveraged the Population Health Outcomes and Information Exchange (PHOENIX [7]) program at Wayne State University to "hotspot" areas with heightened social vulnerability. Third, as the pandemic progressed, we expanded our MHU program to include additional preventive services in hopes of mitigating the impact of COVID-19 on our local community. Indeed, what began as a mobile SARS-CoV-2 testing strategy evolved into a demonstrated capacity for bringing portable healthcare to populations from disadvantaged communities.

# Methods

We performed a formative process evaluation by applying the Centers for Disease Control and Prevention (CDC) framework for public health program evaluation [8]. Our main objective was to determine the success of processes that were implemented to improve health equity by increasing access to difficult-to-reach populations from areas with heightened social vulnerability. We further examined whether the additional screening identified patients at elevated risk of poor health outcomes. Lastly, we determined whether patients requested and received assistance with referral to social services. The Wayne State University Institutional Review Board (IRB) deemed this work to be not human participant research and thus IRB review was waived (WSU IRB 2020 092).

## Mobile health unit program development

The Detroit MHU program began as a rapidly deployed drive-through SARS-CoV-2 testing clinic that was housed in temporary canopy tent shelters at two fixed locations; one in Detroit and the other in Dearborn, Michigan during the first wave of the pandemic. As the local health department established its own large-scale drive-through testing site, our team pivoted towards a focus on patients from socially vulnerable areas who might lack transportation or otherwise be unable to access such services.

Working with Ford X, Ford Motor Company's incubator program, we began by using stock Ford Transit vans to implement a mobile "drive-to" SARS-CoV-2 testing program. Next, the results of field-testing form and function assessments prompted the Ford X team to fully upfit their Ford Transit van platform to better meet our program needs (Fig 1a–1d). Through

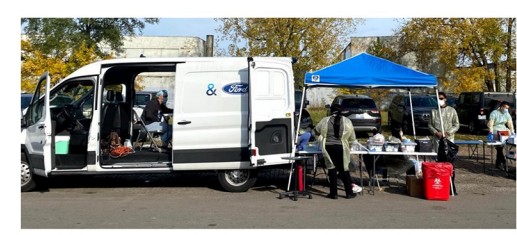
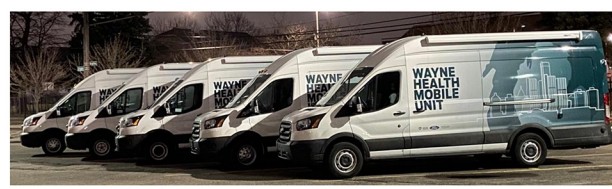
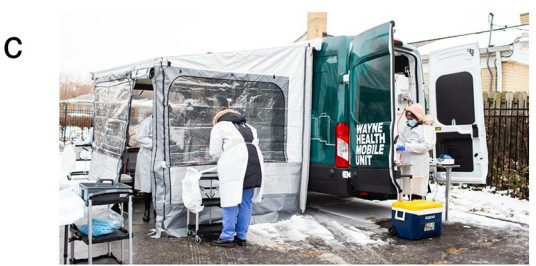
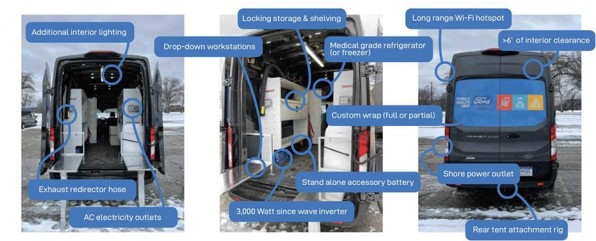

**Fig 1. Mobile health unit design.** Fig 1a-1d show design features of mobile health units. A. Initial Version Based on Stock Ford Transit Platform. B. Upfitted Fleet With Custom Wrap. C. Upfitted Vehicle With Built In Side Awning in Use. D. Overview of Upfitted Vehicle Features.

a grant from the Michigan Department of Health and Human Services (MDHHS) and generous support from the philanthropic community, we were able to purchase and deploy five upfitted vehicles. Program funding also covered the cost of personnel and materials.

## Healthcare and social services

At the outset of our efforts, we partnered with *Patient Education Genius* (PEG; Troy, Michigan), to develop a *de novo*, text-message-based, closed loop, HIPAA-compliant system for patient intake, reporting SARS-COV-2 test results, and data sharing. Electronic informed consent for all services was obtained via PEG, with a parent or guardian consenting for minors, during the registration process. Patients initiated SARS-CoV-2 testing encounters via text messages, services were rendered, and test results were delivered via text message sent automatically by the PEG system. Importantly, these lines of communication remain open as nearly all patients consented to future follow-up contact.

Nasopharyngeal swab SARS-CoV-2 testing was offered to symptomatic healthcare workers and first responders beginning on 3/20/20, ten days after the first two cases were reported in Michigan. The mobile "drive-to" program was launched on 4/13/20 and then subsequently revised as of 9/1/20 to include a more intensive data-drive deployment strategy. Additional program expansions involved antibody testing (Abbott Architect IGG), HIV testing, hypertension screening, additional serology testing, linkage to additional healthcare/social services, and most recently the administration of COVID19 vaccinations. The period of observation and number of patients served are reported in the Results section.

## Data-directed MHU deployment

Site selection for MHU deployment was initially based on publicly available COVID-19 data. "Hot spots" were identified in the surrounding seven counties by calculating SARS-CoV-2 positivity rates per 100,000 people using five-year population estimates from the 2018 US Census American Community Survey. In the absence of open machine-readable data sharing, case

data were aggregated from the public dashboards of Local Health Departments (LHDs) via application program interfaces (APIs). Later as the MHU program expanded, we increasingly relied on real-time data collected onsite.

We began using data provided by the PHOENIX program to target high-risk communities based on emerging evidence that patients from areas with increased social vulnerability might disproportionately suffer adverse COVID-19 outcomes [9]. Our primary measure of community social vulnerability is provided by the CDC Social Vulnerability Index (SVI) [10]. The CDC SVI ranks census tracts on fifteen social factors (e.g., unemployment, minority status, and disability) that are subclassified under four themes: i) Socioeconomic, ii) Household Composition and Disability, iii) Minority Status and Language, and iv) Housing Type and Transportation. We used the SVI for deployment purposes to identify Census Tracts with "racially concentrated poverty" (>40% poverty, >50% non-white), but we also considered chronic disease burden estimates provided by the CDC 500 Cities project [11]. We chose to consider information about chronic disease based on evidence of comorbid cardiometabolic disorders in COVID19, and because neighborhood disadvantage might play a role in the pathogenesis of atherosclerotic/cardiovascular disease-related events [12].

### Statistical evaluation

We performed a formative program evaluation to determine whether our MHU model was able to access people from areas with increased social vulnerability. Patients were classified based on residence in census tracts that received bottom, middle or top quartile CDC SVI rankings. The Chi-Square test and the Cochrane Armitage test were used to test for differences in the proportion of patients from communities with top quartile SVI rankings during three phases of program implementation. Statistical significance was defined with alpha set at 5%. Hypothesis testing was performed using SAS V9.4 (Cary, NC).

## Results

From 3/20/20-3/24/21, the program reached a total of 32,523 people, through 510 total events (491 testing, 19 vaccination) conducted with 218 unique community partners. The median (interquartile range) age was 48 (33–60) years; children comprised 11% of the cohort. Of patients who reported sex, a small majority was female (58%, n = 7339). The most frequently reported race/ethnic category was Black or African American (43%, n = 3395), followed by White (19.3%, n = 1519), Middle Eastern/North African (13%, n = 1059), Hispanic/Latino (12%, n = 964), Asian (6%, n = 472), Multiracial (3%, n = 248) and Native Hawaiian/Pacific Islander (0.2%, n = 14). Among the 11,088 patients who self-reported their medical history, 28% had one or more chronic health conditions.

### SARS-CoV-2 testing

Fig 2 displays the MHU locations overlaid on background SARS-CoV-2 prevalence rates during the observation period (Fig 2). Patients from areas with heightened social vulnerability were over-represented among the patient population as a whole; 33% lived in census tracts with top quartile CDC SVI rankings, while only 25% was expected based on the frequency distribution in the general population. The proportion of patients who resided in communities with top quartile SVI scores increased significantly during the observation period. At the outset during location-based "drive-through" SARS-CoV-2 testing (3/20/20-3/24/21), 25% of the patients were from communities with increased vulnerability. After pivoting to a mobile platform, from 4/13/20-to-8/31/20 the fraction of patients from vulnerable areas increased to 27% (p = 0.01). The adoption of a data-driven deployment strategy resulted in further

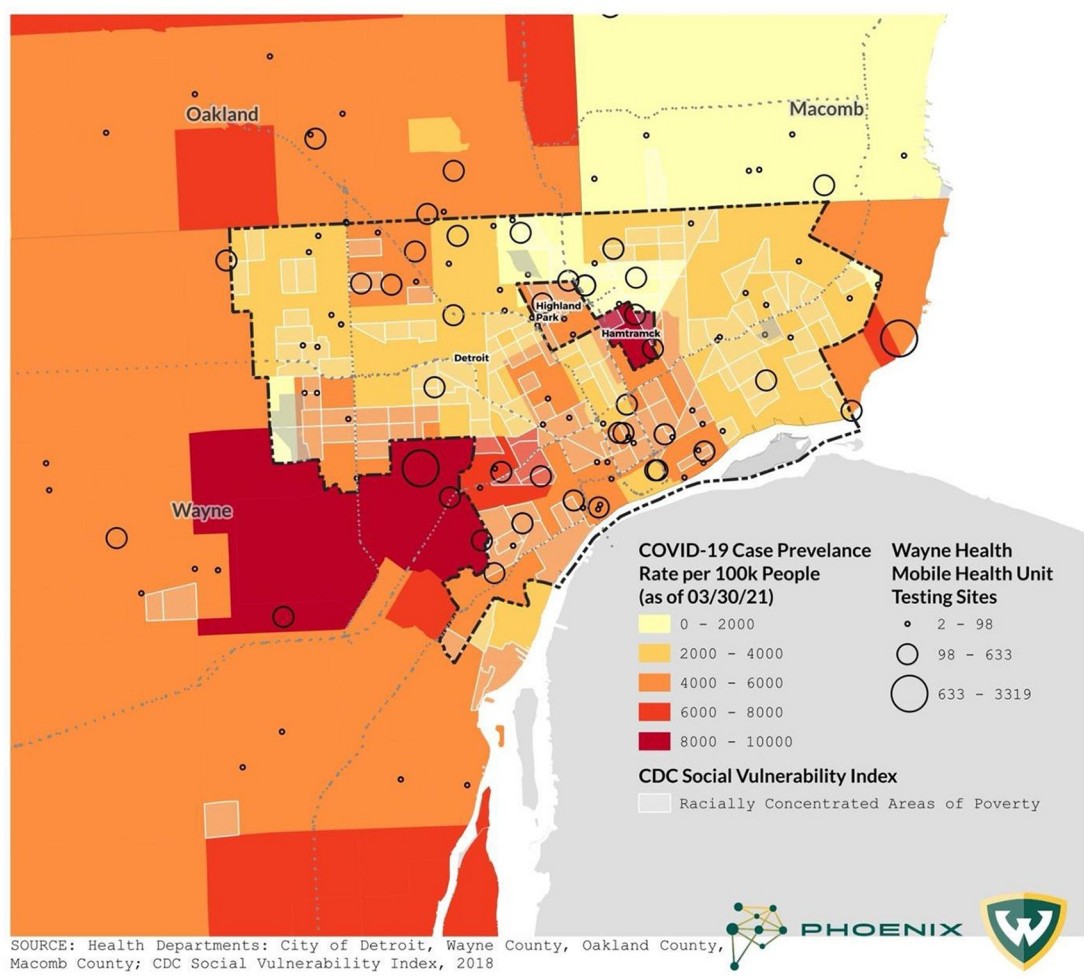

**Fig 2. COVID-19 case rate and mobile health unit testing sites.**

improvement; from 9/1/20-to-3/24/21, 41% of the patients who sought services from the MHUs lived in communities with top quartile SVI scores (Cochrane Armitage test for trend across the three periods, p<0.001).

Patients from areas with increased vulnerability tested positive for SARS-CoV-2 more frequently than patients whose community did not receive a top quartile SVI ranking; however, the pattern was detected only after the data-driven deployment strategy was implemented and background proportion of patients from disadvantaged areas increased (Fig 3).

## Additional services

A timeline of additional program services and numbers of patients reached is provided in Table 1. Through 3/24/21, n = 1,837 patients have received social services assistance (Table 2). The most common request for social services was for food resources (n = 653, 36%). No new cases were identified by HIV testing. Of patients screened for hypertension, nearly half had elevated (>130 mm Hg) systolic blood pressure (46.4%). Fifty-five patients requested linkage to a primary care provider. A total of 4,605 patients received at least one dose of a COVID-19 vaccine from the MHUs.

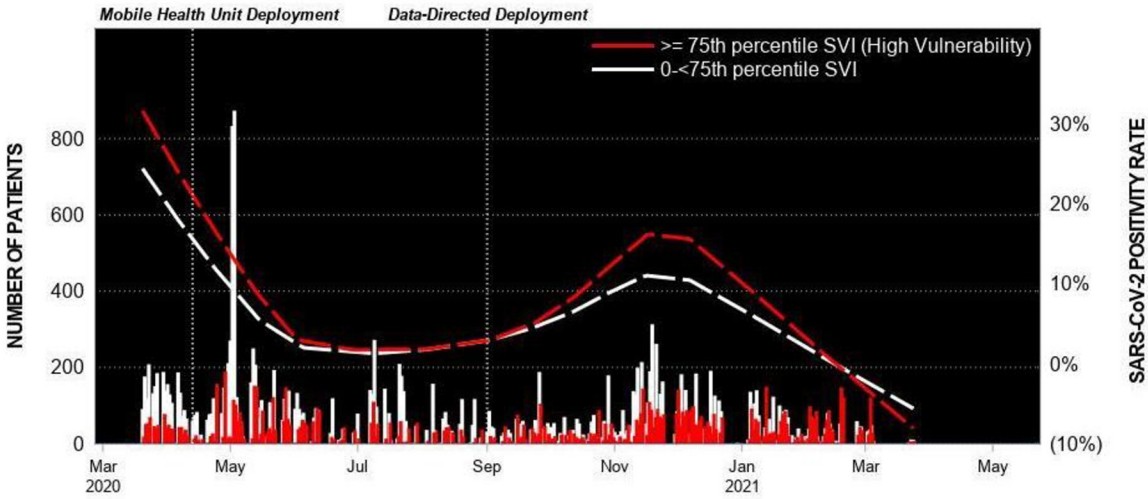

**Fig 3. Mobile health unit testing encounters and SARS-CoV-2 positivity rate.** Positivity rate by residence in an area with top quartile CDC Social Vulnerability Index rankings.

## Discussion

While MHUs have been used in various forms for decades, we developed a novel platform by, i) partnering with community stakeholders to develop custom vehicles, ii) innovating a mobile phone-based application for HIPAA-compliant patient intake and correspondence, and iii) implementing a data-driven deployment strategy. Our experience demonstrates the capacity for MHUs not only to deliver SARS-CoV-2 testing to difficult-to-reach populations, but also to provide preventive healthcare and social services coordination in vulnerable communities. What began as a program focused on providing SARS-CoV-2 testing, evolved into a portable population healthcare delivery system that can address a multitude of community needs.

Importantly, by adopting a data-driven deployment strategy, we further demonstrated that a combination of publicly available information and real time clinical data can be used effectively to increase MHU access to patients from vulnerable communities. Our findings are consistent with pre-SARS-CoV-2 evidence that supports the utility of geospatial solutions as a means to improve crisis response and community resilience [13].

We feel that MHUs will be particularly useful in Michigan, where telehealth uptake is the lowest in the nation during the first year of the novel coronavirus pandemic (15.1% of visits) [14]. Fewer overall clinic visits in-turn resulted in a 50% decline in routine blood pressure

**Table 1. Number of patients served overall and by service type.**

| Service | Start Date | Patients Served (N) |
|---|---|---|
| SARS-CoV-2 Nasal Swab Diagnostic Testing | 3/20/2020 | 29,406 |
| SARS-CoV-2 IGG Antibody Testing | 4/28/2020 | 11,654 |
| HIV Testing | 5/19/2020 | 400 |
| Hypertension Screening | 6/6/2020 | 896 |
| Other Serology Testing (A1c and lipid panel) | 9/26/2020 | 565 |
| Linkage to Care for Social and Medical Services | 10/1/2020 | 1,837 |
| COVID-19 Vaccinations | 3/15/2021 | 4,605 |
| **Total Encounters** | | *32,523* |

**Table 2. Number of social service referrals, follow up attempts and completions ᵃ.**

| Referral Category | Total Assisted | # Attempted Follow Ups | # Completed Follow Ups |
|---|---|---|---|
| Number of individuals assisted with social service referrals onsite | 1837 | 1500 | 822 |
| Food Assistance | 653 | 545 | 321 |
| Public Benefits Assistance | 400 | 285 | 152 |
| Unemployment Assistance | 308 | 256 | 142 |
| Health Insurance Navigation | 176 | 142 | 82 |
| Utility Assistance | 50 | 50 | 50 |
| Voter Registration | 39 | 33 | 22 |
| PCP Referral | 25 | 25 | 25 |
| Transportation Assistance | 5 | 5 | 5 |

ᵃ some patients received more than one service referral.

assessments and a 37% decline in cholesterol screening in 2020 compared to 2018–2019. This is particularly concerning in Detroit, given the high prevalence of hypertension, kidney problems and other chronic diseases [11]. Indeed, our finding that nearly half of the patients served by our program had elevated systolic blood pressure is concerning and indicative of the need for outreach efforts that extend beyond COVID-19 itself.

As indicated by our connectivity with over 200 community partners, we've had broad support for our program from the outset and such partnerships, along with legislation that evolved during the COVID-19 pandemic, helped foster our efforts. Based on our experience, we feel that our MHU model can serve as a mechanism to reduce risk in difficult to reach populations and warrants further investigation as a potentially reimbursable healthcare delivery model. We envision the possibility of a nationwide mobile health corps that is deployed to improve health equity by filling gaps in primary care and chronic disease management in vulnerable areas. Evidence that supports our view comes from previous studies that reported considerable return on investment under similar contexts prior to [15, 16] and during the pandemic [17].

## Strengths and limitations

The major strength of this evaluation is that we applied the CDC Framework for Public Health Program Evaluation to examine whether our processes met the desired objectives. Nevertheless, we do not know if our experience will generalize to similarly vulnerable communities.

## Conclusions

Our descriptive study demonstrates the feasibility of using MHUs to deliver SARS-CoV-2 testing, and additional healthcare/social services, to people from vulnerable areas who are at elevated risk of COVID-19 and its sequelae. Importantly, MHU deployment in at-risk communities created an opportunity to collect information on health and social service deficits that in-turn enabled us to address those very needs.

## Author Contributions

**Conceptualization:** Phillip Levy, Bethany Foster.

**Data curation:** Phillip Levy, Liying Zhang, Steven J. Korzeniewski, Jasmine Criswell, Caitlin O'Brien, Lauren Baird.

**Formal analysis:** Alex B. Hill, Liying Zhang.

**Funding acquisition:** Phillip Levy.

**Investigation:** Phillip Levy, Bethany Foster, Caitlin O'Brien, Katee Dawood, Lauren Baird.

**Methodology:** Steven J. Korzeniewski, Bethany Foster.

**Project administration:** Phillip Levy, Bethany Foster, Jasmine Criswell, Lauren Baird.

**Resources:** Phillip Levy, Erin McGlynn, Alex B. Hill, Liying Zhang, Steven J. Korzeniewski, Caitlin O'Brien, Katee Dawood, Charles J. Shanley.

**Software:** Alex B. Hill, Steven J. Korzeniewski.

**Supervision:** Phillip Levy, Steven J. Korzeniewski, Charles J. Shanley.

**Visualization:** Phillip Levy, Alex B. Hill.

**Writing – original draft:** Phillip Levy, Erin McGlynn, Alex B. Hill, Liying Zhang, Steven J. Korzeniewski, Jasmine Criswell.

**Writing – review & editing:** Phillip Levy, Erin McGlynn, Alex B. Hill, Liying Zhang, Steven J. Korzeniewski, Bethany Foster, Jasmine Criswell, Caitlin O'Brien, Katee Dawood, Lauren Baird, Charles J. Shanley.

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
